# Post-Fire Natural Regeneration Trends in Bolivia: 2001–2021

Oswaldo Maillard

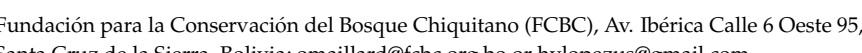

Fundación para la Conservación del Bosque Chiquitano (FCBC), Av. Ibérica Calle 6 Oeste 95, Santa Cruz de la Sierra, Bolivia; omaillard@fcbc.org.bo or hylopezus@gmail.com

**Abstract:** In the last 21 years, Bolivia has recorded a series of thousands of wildfires that impacted an area of 24 million hectares, mainly in the departments of Beni and Santa Cruz. In this sense, identifying trends in the increase of natural vegetation after wildfires is a fundamental step in implementing strategies and public policies to ensure ecosystem recovery. The main objective of this study was to evaluate the spatial trends of the increase and decrease in vegetation affected by wildfires for the whole of Bolivia, for the period 2001–2021, using non-parametric tests, through the analysis of Normalized Difference Vegetation Index (NDVI) remote sensing products. The results indicated that 53.6% of the area showed an increasing trend ($p < 0.05$) and 15.9% of the area showed a decreasing trend ($p < 0.05$). In terms of land cover type, forests were proportionally represented by 18.1% of the areas that showed an increasing trend ($p < 0.05$) and 3.0% of the forests showed a decreasing trend ($p < 0.05$). In contrast, non-forested areas showed an increasing trend of 35.5% and 12.9% showed a decreasing trend ($p < 0.05$). It can be concluded that there is a continuous regeneration process throughout the country.

**Keywords:** fire recurrence; Llanos de Moxos; Chiquitania; NDVI





## 1. Introduction

Bolivia is changing. In recent years it has become one of the countries with the highest forest fire activity in the world [1–3]. Between 2001 and 2020, wildfires affected 1.1 to 9.2 million hectares in the country each year, mainly in the lowlands [4]. The fires of 2019 in the Chiquitania region, in the department of Santa Cruz, had remarkable attention and the repercussions in the public opinion brought the problem of fires to a national debate, capturing the world's attention. Thousands of individuals volunteered to fight the fires and rescue animals, and charity activities were held to raise funds. In addition, several foreign governments contributed financial resources and equipment to help fight the fires and restore the areas affected by the wildfires.

To counteract the environmental impact generated by these wildfires, the Government of the Plurinational State of Bolivia, together with the Autonomous Departmental Government of Santa Cruz, developed the plan for the recovery of areas affected by fires in the department of Santa Cruz with an emphasis on the Chiquitania region, and subsequently prepared a strategic document for the implementation of this plan [5]. Although restoration actions with a passive or active approach in deforested and fire-impacted areas are essential to fulfill Bolivia's commitments to the international objectives of the global climate agenda (the Bonn Challenge, the New York Declaration on Forests, and the UN Decade of Ecosystem Restoration), there are many limitations in the country, mainly due to the scarcity of financial resources, which do not allow the implementation of this plan [5].

While different actions are being promoted for the restoration of the ecosystems impacted by the fires in the Chiquitania region, some of the areas that were in the process of recovery are burning again. This has created a series of questions regarding the capacity of ecosystems to withstand different degrees of disturbance. Over the last 20 years, a series of investigations have been conducted to understand the processes of natural regeneration in microsites at the species level [6] in dry and humid forests in the Chiquitania region

after selective logging [7–12], controlled burns [8,13], and wildfires [10,14–17]. These studies used field sampling plots, and the results show that there are different vegetation regeneration processes in burned sites, suggesting that the dry forests of the Chiquitania region are adapted to fire, but not humid forests. Dry ecosystems have a high survival rate of trees after a fire and most tree species use sprouting as one of their survival strategies [17]. However, it is still unclear how regeneration processes occur spatially at the landscape scale, in different ecosystems, and with different wildfire recurrences.

Due to its multi-scale and multi-temporal capabilities, remote sensing has shown to be a useful and affordable tool for spatial monitoring of land cover change [18]. This is because it is particularly suitable for quantifying patterns of variation in space and time, and detecting changes caused by both natural and anthropogenic disturbances [19]. Most analyses use satellite-derived spectral indices (e.g., Normalized Difference Vegetation Index, NDVI), which can serve as indicators to detect trends in vegetation activity change [20]. Furthermore, remote sensing is crucial for identifying fire regimes [21], assessing the destruction caused by fire to forest ecosystems [3], and keeping track of post-fire vegetation regeneration [22]. The development of prioritized restoration plans depends on this information [23]. However, spatial analyses of regeneration at the national scale are scarce and are needed to develop public policies and actions. In the last three years, increased research has been conducted to evaluate post-fire natural regeneration in the Chiquitania region using satellite imagery in combination with field plots [5,24,25]. However, until now, no study has been conducted to evaluate post-fire natural regeneration trends for the entirety of Bolivia.

The primary goal of this study was to determine the spatial trend pattern of natural vegetation in response to wildfires through a temporal NDVI analysis, and its implication on natural regeneration occurring spontaneously in Bolivia during the period 2001–2021. Understanding the post-fire regeneration processes is crucial for allocating resources more effectively to socioeconomic and environmental policies.

## 2. Materials and Methods

### 2.1. Study Area

Bolivia is administratively and politically conformed by nine departments (Beni, Chuquisaca, Cochabamba, La Paz, Oruro, Pando, Potosí, Santa Cruz, and Tarija). In addition, this country is one of great contrasts, ranging in altitude, climate, and vegetation from the tops of high mountains to tropical forests. Geomorphologically, there is a high Andean region in the western part of the country, which is crossed by the Cordillera Occidental and Cordillera Oriental, which frame a central depression known as the Altiplano (between 3400 and 4000 m a.s.l.). In the central region there is a group of parallel mountain ranges known as the Sub-Andean belt, and to the east are the extensive plains of the lowland region (<1000 m) which occupy most of the territory with the presence of low to medium altitude hills. Among the iconic lowland regions are the Llanos de Moxos in Beni and Chiquitania in Santa Cruz (Figure 1).

Due to this complex topography, Bolivia concentrates 12 important ecoregions, each with a high diversity of vegetation formations [26]. Seven ecoregions are located in the Andean and sub-Andean regions (Figure 1): Northern Puna, Southern Puna, Inter-Andean Dry Valleys, Prepuna, Yungas, Boliviano-Tucumano Forest, and Chaco Serrano [26]. Five ecoregions are located in the lowland's region (Figure 1): Southwest Amazonia, Chiquitano Dry Forest, Chaco, Cerrado, and the Seasonally Flooded Savanna (Pantanal and Llanos de Moxos) [26]. In the last two decades, a large proportion of the ecoregions have been impacted by fires [4], especially in the lowlands (Figures 1 and 2). Savannas are proportionally the ecosystems most affected annually by fires (42–62%) [4]. After fires, some of these ecosystems regenerate naturally (Figure 2) [14–17].

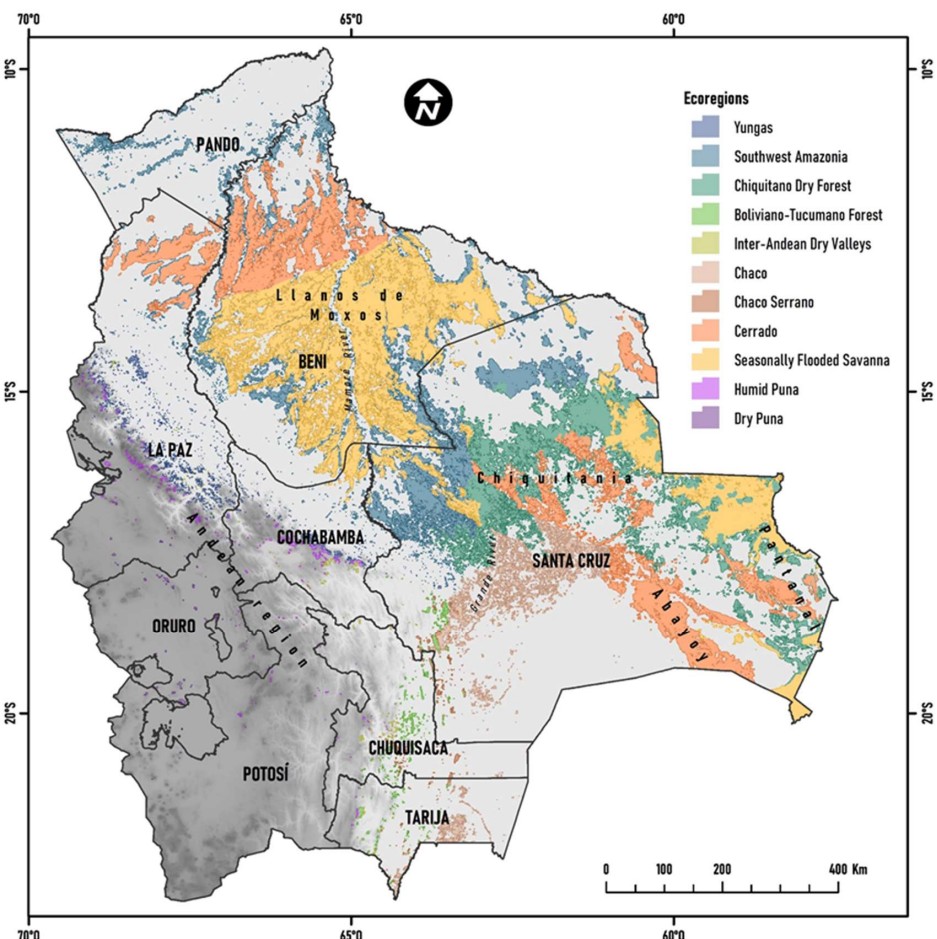

**Figure 1.** Wildfire scars (2001–2021) relation to the ecoregions of Bolivia and areas of interest mentioned in this research.

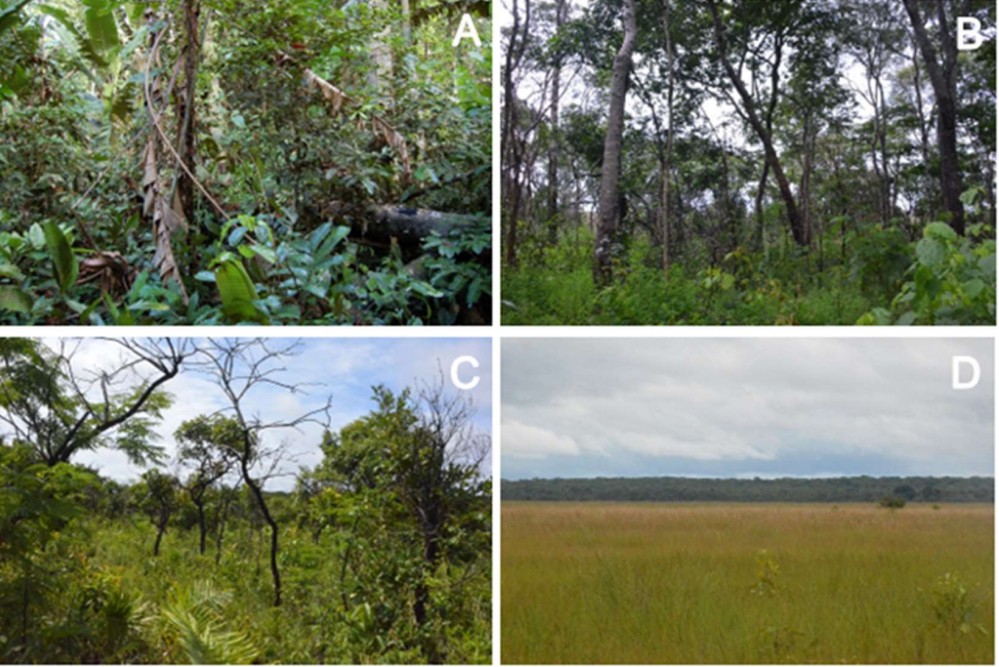

**Figure 2.** Ecoregions impacted by wildfires in Bolivia in the process of post-fire natural regeneration: (**A**) Southwest Amazonia, (**B**) Chiquitano Dry Forest, (**C**) Cerrado, and (**D**) Seasonally Flooded Savanna. Photos: Oswaldo Maillard.

## 2.2. Datasets

A total of three remote sensing datasets were used in this study, including burned areas, vegetation conditions, and land use and land cover.

### 2.2.1. Burned Areas

To determine the area impacted by wildfires in Bolivia, I used the monthly product MCD64A1 ver. 6 from the combination of Terra and Aqua (MODIS) satellites, with a spatial resolution of 500 m (https://lpdaac.usgs.gov/products/mcd64a1v006, accessed on 21 February 2022). This product is made using an automated system that recognizes quick patterns in surface reflectivity time-series variations detected by MODIS sensors [27]. This system also reduces the size of unmapped areas in areas with heavy cloud cover and improves the ability to detect small burns [27]. While dense clouds and smoke limit the accuracy of burned area products, MCD64A1 shows the best performance compared to other burned area products [28,29].

### 2.2.2. Normalized Difference Vegetation Index

To determine natural regeneration, I used the MOD13A2 Version 6 product (https://lpdaac.usgs.gov/products/mod13a2v006, accessed on 21 February 2022), which provides Normalized Difference Vegetation Index (NDVI) values every 16 days. MOD13A2 was designed to provide spatially and temporally consistent comparisons of vegetation conditions and was processed from the MODIS L3 daily surface reflectance product corrected for the effects of atmospheric gases, thin cirrus clouds, and aerosols [30]. In recent decades, MOD13A2 data have been used to quantify vegetation activity and detect vegetation dynamics in many biological communities [31–33].

### 2.2.3. Land Cover Type

I evaluated post-fire natural regeneration in Bolivian ecosystems, using the MODIS Land Cover Type Product MCD12Q1.006 (https://lpdaac.usgs.gov/products/mcd12q1v006, accessed on 21 February 2022), a series of global land cover maps with yearly intervals and 500 m spatial resolution from 2001 to 2020. Supervised classifications of MODIS Terra and Aqua reflectance data were used to create this product [34]. I used the University of Maryland (UMD) classification (Type 2). The MCD12Q1.006 product was used in this study because its spatial resolution integrates with other MODIS products, the classification covers all of Bolivia, and it has a broad time scale (20 years).

## 2.3. Data Processing and Statistical Analyses

Using the cloud computing platform Google Earth Engine (GEE, [35]), I developed a script to generate annual cumulative images of the NDVI time series and detect trend changes between 2001 and 2021. To detect these trends, I employed the Mann-Kendall statistical test [36]. The Mann-Kendall test is a non-parametric test used mainly to determine statistically significant monotonic trends in a time series of Z-scores [37]. Each time step was compared to the previous step to evaluate the temporal trends of each pixel. The sum of all the pairwise signs is what is known as the Mann-Kendall trend [37]. The result was +1 if the Z score at the second time step was greater than the Z score at the first time step, −1 if the opposite is true, and zero if they were both equal. Over the 21-year period, each pair of time steps was compared, obtaining the Mann-Kendall statistic with the associated trend Z-score and *p*-value for each pixel. To facilitate a direct comparison between different regions worldwide, I presented results using four categories of trends: positive and significant ($p < 0.05$), positive and non-significant ($p > 0.05$), negative and non-significant ($p > 0.05$), and negative and significant ($p < 0.05$). Positive values indicate an increasing trend in vegetation cover and negative values indicate a decreasing trend.

I also developed a script in GEE to generate and download the cumulative burned area scars for each year (2001–2021) based on the MCD64A1 product, and another script for the annual MCD12Q1.006 land cover type product (2001–2020). In ArcMap 10.8, I

processed all annual fire scars to obtain a single binary image and intersected it with the mapping of Bolivian departmental boundaries, and subsequently calculated the areas. I identified fire recurrences by year and grouped them by ranges (1, 2–4, 5–7, 8–10, and >10). In addition, I used a binary image of fire scars to extract the values of the land cover type layer. In addition, the product MCD12Q1.006 was reclassified into a new map of forest and non-forest cover, comparing the years 2001 and 2020 to detect areas of land use change and avoid classification errors. I used the binary image, the layer of burned area recurrences, and the type of cover impacted by fire, forest and non-forest cover, and intersected them with the values obtained in the Mann-Kendall test. Finally, I classified the four trends of change and their levels of statistical significance ($p < 0.05$), and the percentages of the trends of change were obtained.

## 3. Results

### 3.1. Regeneration Trends by Region

The results of natural regeneration trends over 21 years of fire events in Bolivia show diverse spatial variations. The green colors in Figure 3 represent increasing and significant trends ($p < 0.05$), while the red colors represent decreasing and significant trends ($p < 0.05$). The NDVI in 74.7% (53.6% significant, $p < 0.05$) of the study area showed an increasing trend and 25.3% (15.9% significant, $p < 0.05$) of the area presents a decreasing trend (Table 1, Figure 2). The departments in the country showing increasing and significant ($p < 0.05$) trends were mainly Beni (30.2%), Santa Cruz (19.5%), and La Paz (2.4%). However, significant ($p < 0.05$) decreasing trends were also evident in Santa Cruz (11.9%) and Beni (2.6%).

**Table 1.** Post-fire natural regeneration trends for the nine administrative departments of Bolivia, based on the Mann-Kendall test (increasing, decreasing) and statistically significance values ($p < 0.05$, $p > 0.05$). The data are presented in percentages in relation to the total burned area in the period 2001–2021.

| Departament | Burned Areas (ha) | Increasing/ Significant ($p < 0.05$) | Increasing/ Non-Significant ($p > 0.05$) | Decreasing/ Non-Significant ($p > 0.05$) | Decreasing/ Significant ($p < 0.05$) | Total |
|---|---|---|---|---|---|---|
| Beni | 10,121,319 | 30.2 | 6.7 | 2.7 | 2.6 | 42.2 |
| Chuquisaca | 99,229 | 0.3 | 0.1 | 0.0 | 0.0 | 0.4 |
| Cochabamba | 292,685 | 0.6 | 0.2 | 0.1 | 0.2 | 1.1 |
| La Paz | 1,142,139 | 2.4 | 1.2 | 0.5 | 0.6 | 4.7 |
| Oruro | 18,189 | 0.0 | 0.0 | 0.0 | 0.0 | 0.0 |
| Pando | 264,624 | 0.4 | 0.2 | 0.1 | 0.3 | 1.0 |
| Potosí | 6404 | 0.0 | 0.0 | 0.0 | 0.0 | 0.0 |
| Santa Cruz | 12,060,676 | 19.5 | 12.6 | 5.9 | 11.9 | 49.9 |
| Tarija | 175,302 | 0.2 | 0.1 | 0.1 | 0.3 | 0.7 |
| Total | 24,180,566 | 53.6 | 21.1 | 9.4 | 15.9 | 100 |

In the Department of Beni, in the regions of the Mamoré River, known as the Llanos de Moxos, an extensive savannah that is seasonally flooded (Figure 4), it is observed that the trends of NDVI changes are mainly significantly increasing (green) with some scattered areas as decreasing trends (red). However, in the Department of Santa Cruz, west of the Grande River, an area characterized by a mosaic of agricultural and livestock activities, the trends of change are mainly decreasing, but there are scattered areas with increasing trends that are known to be from forested and non-forested formations (Figure 4). In addition, in Concepción, San Matías, and San José-Roboré areas, the increasing trends were greater in forested areas, shrublands, and pastures, but decreased in deforested or non-forested areas (Figure 4).

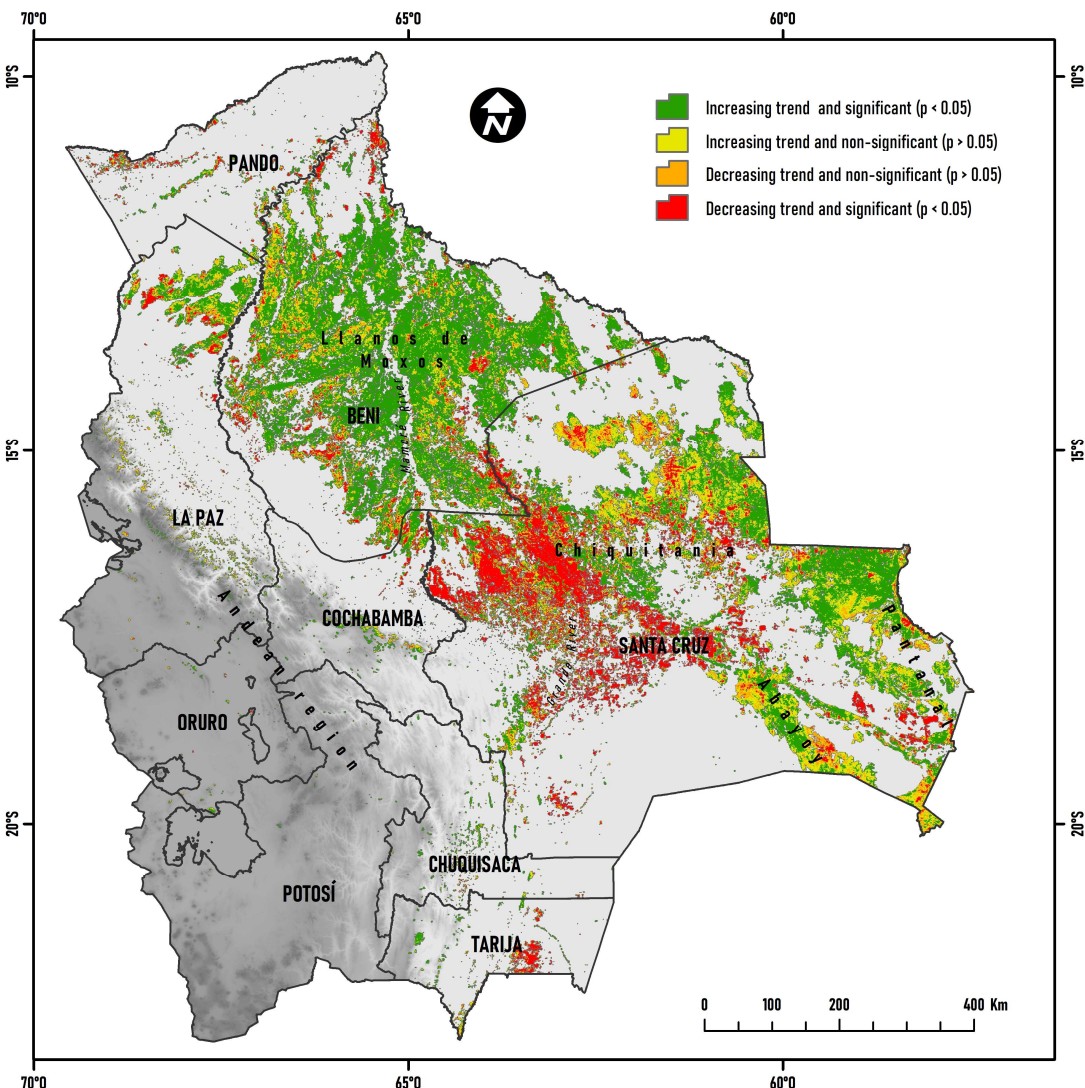

**Figure 3.** Spatial trend of post-fire natural cover increases or decreases in Bolivia (2001–2021) based on the Mann-Kendall test and statistically significance values ($p < 0.05$, $p > 0.05$).

### 3.2. Regeneration Trends by Land Cover Type

In terms of land cover type, proportionally in relation to the areas impacted by fires, forests were represented by 28.0% (18.1% significant, $p < 0.05$) in areas that showed an increasing trend, and 6.8% (3.0% significant, $p < 0.05$) of forests showed a decreasing trend (Table 2). In the forested areas with significant statistical trends ($p < 0.05$), these are mainly concentrated in the Evergreen Broadleaf Forests (9.2%) and Deciduous Broadleaf Forests (8.6%), while in some areas there were decreasing trends for Evergreen Broadleaved Forests (2.2%) (Table 2). In contrast, in the non-forested areas (65.3%) significant increasing trends ($p < 0.05$) were observed in Savannas (23.0%) and Grasslands (6.9%), as well as decreasing trend for Grasslands (4.6%), Savannas (3.7%), Croplands (2.6%), and Woody Savannas (1.8%) (Table 2).

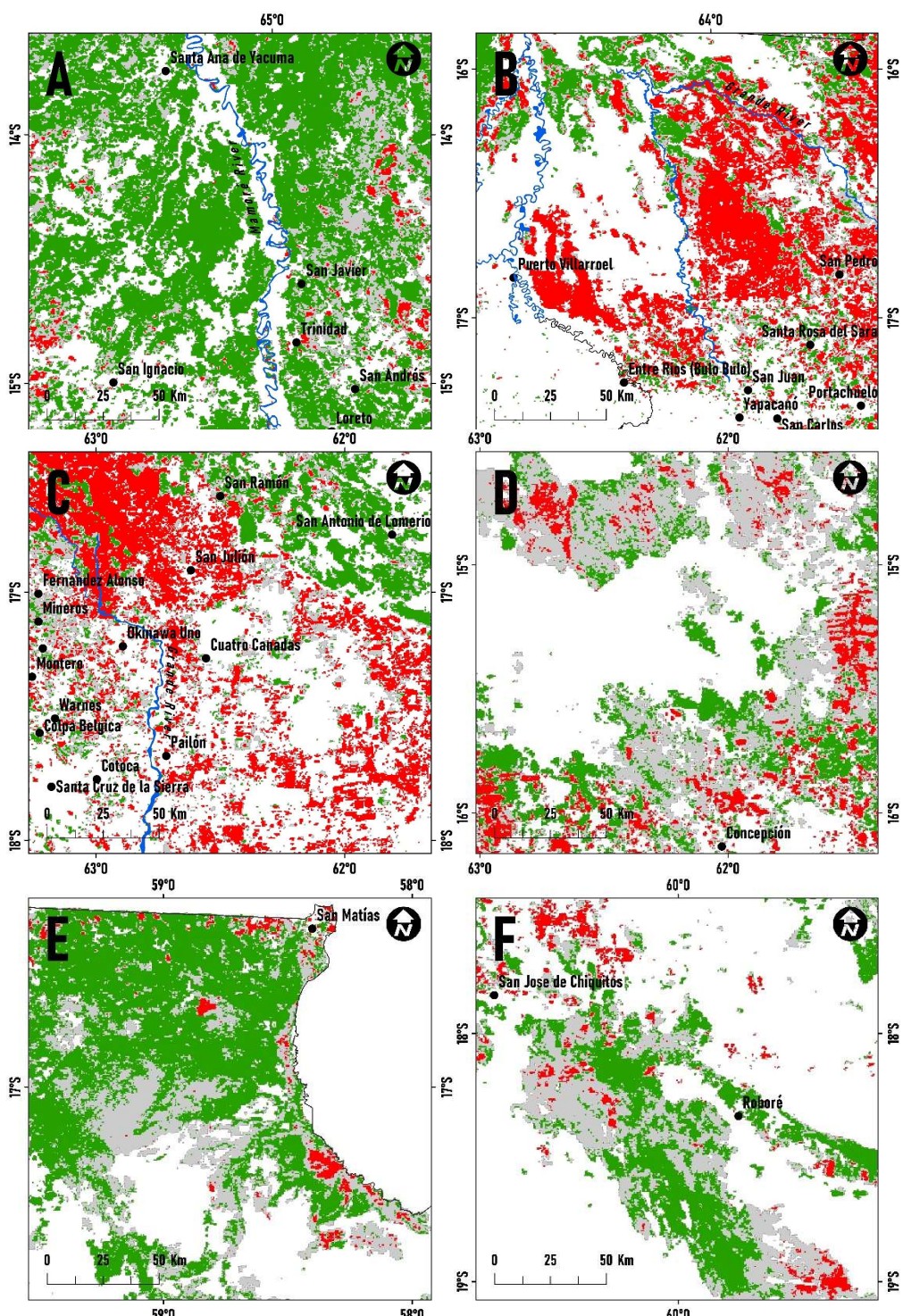

**Figure 4.** Visualization of areas with significant increasing (green) and decreasing (red) trends in (**A**) Mamoré River, (**B**) West of the Grande River, (**C**) East of the Grande River, (**D**) Concepción, (**E**) San Matías, and (**F**) San José-Roboré. Areas in gray have no significant trend.

**Table 2.** Trends of natural regeneration in fire scar area for land cover types in Bolivia, based on the Mann-Kendall test (increasing, decreasing) and statistically significance values ($p < 0.05$, $p > 0.05$). The data are presented in percentages in relation to the total burned area in the period 2001–2021.

| Land Cover Type | Burned Áreas (ha) | Increasing/ Significant ($p < 0.05$) | Increasing/ Non-Significant ($p > 0.05$) | Decreasing/ Non-Significant ($p > 0.05$) | Decreasing/ Significant ($p < 0.05$) | Total |
|---|---|---|---|---|---|---|
| Evergreen Needleleaf Forests | 4508 | 0.0 | 0.0 | 0.0 | 0.0 | 0.0 |
| Evergreen Broadleaf Forests | 4,422,915 | 9.2 | 4.4 | 2.0 | 2.2 | 17.7 |
| Deciduous Needleleaf Forests | 25 | 0.0 | 0.0 | - | - | 0.0 |
| Deciduous Broadleaf Forests | 3,999,299 | 8.6 | 5.4 | 1.7 | 0.8 | 16.6 |
| Mixed Forests | 96,223 | 0.3 | 0.1 | 0.0 | 0.0 | 0.4 |
| Closed Shrublands | 1500 | 0.0 | 0.0 | 0.0 | 0.0 | 0.0 |
| Open Shrublands | 110,493 | 0.3 | 0.1 | 0.0 | 0.0 | 0.4 |
| Woody Savannas | 2,356,589 | 4.6 | 2.3 | 1.1 | 1.8 | 9.8 |
| Savannas | 8,079,829 | 23.0 | 5.1 | 2.4 | 3.7 | 34.2 |
| Grasslands | 3,884,840 | 6.9 | 3.2 | 1.5 | 4.6 | 16.2 |
| Permanent Wetlands | 143,498 | 0.3 | 0.1 | 0.1 | 0.1 | 0.6 |
| Croplands | 1,002,063 | 0.3 | 0.4 | 0.6 | 2.6 | 3.9 |
| Urban and Built-up Lands | 8408 | 0.0 | 0.0 | 0.0 | 0.0 | 0.0 |
| Cropland/Natural Vegetation Mosaics | 57,000 | 0.1 | 0.0 | 0.0 | 0.1 | 0.2 |
| Non-Vegetated Lands | 13,377 | 0.0 | 0.0 | 0.0 | 0.0 | 0.0 |
| Total | 24,180,566 | 53.6 | 21.1 | 9.4 | 15.9 | 100 |

### 3.3. Regeneration Trends Based on Wildfire Reoccurrence

I found that fire recurrences between 2001 and 2021 are spatially represented mainly by sites that burned between 2 to 4 years (39.3%), followed by those that burned only once (35.7%), between 5–7 (15.8%), 8–10 (6.6%), and more than 10 years (2.7%). In forested areas, the areas that proportionally presented the largest areas with an increasing trend were located in areas that burned in a single year (41.7%) and 2–4 years (40.6%), whereas those with decreasing trends were represented by 52.0% in sites that burned only once and 44.4% between 2 to 4 years (Figure 5). In non-forested areas, the highest proportions with increasing trends were found in sites burned 2–4 years (42.4%), 5–7 years (23.4%), and one year (20.6%), while decreasing trends were found in sites burned 2–4 times (40.6%) and once (39.2%) (Figure 5).

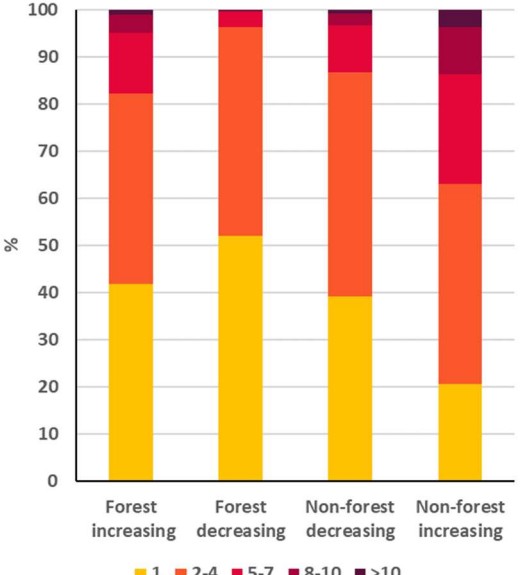

**Figure 5.** Proportion of the increase or decrease in forest and non-forest cover in areas with different ranges of annual forest fire recurrences in Bolivia (1, 2–4, 5–7, 8–10, and >10), according to the Mann-Kendall test and statistically significance values.

## 4. Discussion

This study analyzed the long-term post-fire natural regeneration trends over the entirety of Bolivia. In 21 years, more than 24 million hectares of the country have been impacted by wildfire, mainly in the lowland region. Although wildfire dynamics in Bolivia have been extensively evaluated [1,2,4,38–40], very few studies have been conducted so far to analyze post-fire regeneration processes using remote sensing [5,24] and a complete spatial analysis at the national level was not previously available. The results of this study indicate that more than half (54%) of the burned areas in Bolivia show a statistically significant and increasing trend, mainly in those located in the department of Beni, in Llanos de Moxos. However, almost 16% of the burned areas in Bolivia have a decreasing and significant trend, mainly located in the department of Santa Cruz, in landscapes characterized by forests, scrublands, and agricultural, or livestock areas. These data are fundamental to be able to initiate the approach of any publication policy strategy for regeneration actions.

In many dry and humid ecosystems, including savannas and forests, fire represents a natural ecological factor [41,42]. Although it is known that among the direct causes of forest degradation in Bolivia are wildfires [43], which occur every year and are mainly of anthropogenic origin [38,39,43,44]. In the Andean region, the most important factors in the process of forest (*Polylepis*) destruction are wildfires [45]. In the lowlands, in other ecosystems such as humid forests, periodic fire may play an important role in successional development [42]. The results show that at the land cover type level, the forests that presented spatially the highest proportions of significantly increasing and decreasing trends were the Evergreen Broadleaf Forests, in the Southwest Amazonia ecoregion [26]. Amazonian forests have a high capacity for regeneration through secondary succession [46]. The results presented also showed a representative proportion of the increasing trend for Deciduous Broadleaf Forests, known as the Chiquitano Dry Forest [26]. Recent studies based on field sampling have indicated that forests, especially dry forests, have a high capacity to regenerate after wildfires [15]. In the Chiquitano Forest many species are adapted to fire by presenting thick and corky bark with relatively slow growth and very high wood densities [47]. For this reason, the results show that, in areas with fire disturbance caused in many forested areas, a sharp drop in NDVI values is due to vegetation degradation followed by a rapid recovery of herbaceous and deciduous vegetation, leading to strong increases in NDVI, something that has been observed previously for the Chiquitania region [5]. It is for this reason that it should be considered that forests are not necessarily lost during wildfires, as shown in some global analyses [3], as they may be degraded or in any case are regenerating again [17].

However, in non-forested areas, the results show that Savannas and Grasslands, located in the Cerrado and Seasonally Flooded Savanna ecoregions [26], presented spatially the highest proportions of increasing (29.9%) and decreasing (8.3%) trends. Savannas and grasslands landscapes are shaped by annual burning and are part of the ecology of these ecosystems [42,48,49]. Each year fires in non-forested areas are more frequent between August and October, at the end of the dry season, and some may be of natural origin [42]. In the Llanos de Moxos, a seasonally flooded tropical savanna of more than 160 thousand km$^2$ containing small islands of palm and gallery forests [50], present habitats are shaped by a multitude of abiotic conditions and disturbance factors, including fires [4,49–51]. In addition, the forest islands of Llanos de Moxos are characterized by the presence of fire-resistant species, resulting in rapid natural regeneration [52,53]. Furthermore, in other savanna ecosystems such as Noel Kempff National Park in the western department of Santa Cruz, fire is not only responsible for the formation and maintenance of different habitats, but it also contributes to the heterogeneity of structures and species composition [41]. For example, in Shrubby Savannas and Open Woodland, several species adapt to avoid fire damage and species flourish only after having experienced recent burns [41]. This explains why a large proportion of the non-forested landscapes showed a strong increase in NDVI values. However, some extreme fire events in non-forested formations are still being

evaluated. In the southern San José-Roboré area, where a Cerrado formation known as Abayoy is located, the most intense fire in all of Bolivia in the last 20 years was recorded [4]. Despite the magnitude of the fire producing high tree mortality, high percentages of regrowth were recorded [17], which was confirmed using remote sensing [5].

In addition, this study shows that some areas of MODIS fire scars are located in the area west of the Grande River, which represent sites with expanding deforestation processes [54], dedicated to agricultural and livestock activities. Similarly, other scattered sites with decreasing trends in Chiquitania (Concepción, San Matías, and San José-Roboré), presented areas disturbed by productive activities or by recent wildfires, which led to strong decreases in NDVI values. Therefore, the interpretation of the results should be handled with caution. In Bolivia, fire management is essential because it is the cheapest tool used during chaqueos practices (slash-and-burn agriculture to prepare the land for planting), mechanized agriculture for commercial purposes, and to increase the availability of palatable forage for livestock [26,38,39,43,55]. Generally, many of these burns are out of control and cause fires that impact natural vegetation [26,42,43]. While the vegetation is recovering naturally, in the following years, some of these areas will burn again, resulting in a seemingly endless cycle. As a conservation problem, fires and burns should be addressed holistically, through good fire management practices and the strengthening of management with public policies [39]. But while advancing in good fire practices, the recovery of ecosystems impacted by fire should be ensured through the planning of cost-effective operational strategies and actions, agreed upon by governments (national, departmental, and municipal) and civil society organizations, promoting regeneration and, if necessary, assisted restoration. The efforts of these actions have to focus mainly on the forested areas that presented significantly decreasing trends in this study, as is the case of the Evergreen Broadleaf Forests and Deciduous Broadleaf Forests.

In terms of fire recurrence, the areas that presented spatially the highest proportions were those where fire impacted areas with frequencies between 2 and 4 years and those that burned only once in a 21-year period. In forest formations, the proportions of increasing and decreasing trends were concentrated mainly in 1 and 2–4 years. The increase in the frequency of forest fires in tropical regions generates changes in forest composition, structure, floristic diversity, and economic value [56,57]. Studies conducted in the Chiquitano Forest found that forest regeneration depends on fire severity levels [10,15] but is also related to fire frequency [58]. For non-forested areas, increasing trends are found mainly in 2–4 times, 5–7 times (23.4%), and once, but of decreasing ones in 2–4 times and only once. In the case of fires in non-forested areas, it is known that fire is considered the main regulator of successional dynamics [42,52,53]. In the case of the Cerrado, a high burning frequency can have a negative impact, especially on the mortality of younger trees [17,48]. It is important to continue with further studies to determine the effect of different fire frequencies and intensities on the forest-savanna mosaic, not only in terms of the dynamics between these two ecoregions, but also their respective structure and species composition [50].

## 5. Conclusions

This is the first study to spatially identify trends of vegetation increase and decrease in Bolivia in a long data series. The NDVI remains an essential tool for monitoring the health of the countryside and determining the loss or maintenance of forest cover. The findings of this study have considerable relevance for the ecology and management of fire-affected ecosystems in Bolivia. It was concluded that more than half of the burned areas in the country are in a continuous process of regeneration; however, some sites are in the process of degradation due to recurrent fires. The identification of sites with decreasing trends, especially in forested areas (Evergreen Broadleaf Forests and Deciduous Broadleaf Forests), could qualify the sites for assisted restoration actions.

A limitation of this study was the special resolution used so future studies should use medium resolution images, such as Landsat, with longer time series to quantify and identify small forest areas at a smaller scale and determine more accurately the trends of

increase or decrease in vegetation throughout Bolivia. In addition, these analyses have to be performed in combination with field data.

**Funding:** The publication of the research was made possible thanks to financial support from the Government of Canada under the Knowledge Bases for Restoration-III project (Restauracción).

**Institutional Review Board Statement:** Not applicable.

**Informed Consent Statement:** Not applicable.

**Data Availability Statement:** Not applicable.

**Acknowledgments:** Analyses of post-fire natural regeneration in Bolivia have been developed as part of a series of investigations by the FCBC Chiquitano Dry Forest Observatory. I thank the four reviewers for their insightful comments and suggestions that greatly helped in improving our manuscript.

**Conflicts of Interest:** The author declares no conflict of interest.

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
