# Peer review of "Post-Fire Natural Regeneration Trends in Bolivia: 2001–2021"

_fire, doi:10.3390/fire6010018_

Round 1

Reviewer 1 Report

The introduction needs an improvement For example you can mention also remote sensing limitations and add more references ( line 64, 65). I have to point that introduction needs to be expanded. 
Line 78: “Geomorphologically, these two regions….” Which two regions? Make it clear
Line 222-227: not polite, not well written. Please rephrase 
Line 297-305: are important limitations of the manuscript. Please try it to justify them more or add some words in the introduction. 
Table 1: Totals are not correct 

Author Response

We would like to thank the reviewer for his/her assessment of the manuscript and the constructive comments that he/she has provided. We have made several adjustments to the document based on your suggestions.

Reviewer 2 Report

The manuscript has good structure, but the conclusion should be expanded. Some parts of the discussion could be transferred to the conclusion.

There are typing errors in the manuscript (for example, rows 67, 152). The authors should check the whole text. They should pay special attention on the reference list, since there are many mistakes (different styles).

I believe that the manuscript, after the changes made, should be accepted for publication.

Author Response

Thank you for your positive evaluation and constructive comments on the manuscript.  Here I include a detailed response to all your comments.

Reviewer 3 Report

1.I counter-recommend about including lists of terms in the text as done in lines 83-86. They could be included in the footnote in order to keep reading fluent;
2.lines 98-102: please discuss omission by the MCD64A1 satellite product as it is known to underestimate burned are due to cloud coverage. How was this mitigated must be informed to the reader;
3.lines 112-118: why was Landsat whose resolution is much higher, not used? Please inform the reader at least in a footnote;
4.line 140-143: the phrase needs to be rewritten especially lines 141-142. Also what does misclassification exactly means? Please add a short explanation within parentheses;
5.Section 2.3: the reader must be informed with more detail how the bi-annual trend patterns were converted into a dominant trend assessment by the non-parametric techique used. For instance, was some threshold used, such as, e.g., if positive trend in at least 50% of time, then the pixel was classified as exhibiting a positive trend?
6.Table 2: add one last column with the percent of the change categories (i.e., the share that remained unchanged), so that the reader will know that the five columns sum to 100%, both in each row and across all rows and columns.
7.Discussion: there's need of making clear how results are useful in terms of regeneration management by Bolivian national and subnational governments. The mention to fire usage in lines 264-280 do not refer to regeneration management, which is the key policy front that results may contribute for. How can government assist regeneration, accelerating it? Which are the priority locations and forest types for such governmental action?

Author Response

Thank you for your positive evaluation and constructive comments on the manuscript. These comments have contributed significantly to improving the readability and clarity of the manuscript. I made several adjustments to the document based on your suggestions.

Reviewer 4 Report

Dear Authors,

the manuscript "Post-fire natural regeneration trends in Bolivia: 2001-2021" intends to evaluate the spatial trends of variation of vegetation structures induced by fires in Bolivia in the time period 2001-2021.

Although an adequate and up-to-date literature was introduced at the beginning to define clear objectives and then taken up in the discussions, I believe that the manuscript in question is a mere exercise in spatial analysis of the temporal variation of the NDVI vegetation index. In the title and throughout the document, an attempt is made to equate the temporal and spatial variations of NDVi to presumed vegetational structures, renewal and/or degradation, without there being any connection between the NDVI data and the truths on the ground. An extension of the value of the vegetation indices to the reality of the associated vegetation dynamics is only possible through a comparison of sample data obtainable from appropriate field vegetation surveys.

Author Response

R: A: Dear reviewer,

Thank you very much for your valuable comment, it is valid and quite appropriate.

Indeed I performed a spatial analysis of the temporal trend using a long series of NDVI data to identify the increase or decrease of vegetation for each department, different types of ecosystems and an evaluation according to fire recurrences. My objective was not to identify how the regeneration processes are at each site, that will be achieved through further field studies, this relationship is mentioned in the discussion. Doing a field-based analysis for 24 million hectares is not realistic and that is why these analyses are used over a long period of time. This is a first step for further studies. In the conclusion we mention this.

My kind answer is that I will not make changes in the title or profound changes in the document. I did make several changes based on the other three reviewers. As you will see, this helped to greatly improve the writing and organization 
